# A retrospective analysis of pathogen profile, antimicrobial resistance and mortality in neonatal hospital-acquired bloodstream infections from 2009–2018 at Tygerberg Hospital, South Africa

**Kessendri Reddy**[1]*, **Adrie Bekker**[2], **Andrew C. Whitelaw**[1], **Tonya M. Esterhuizen**[3], **Angela Dramowski**[4]

1 Division of Medical Microbiology and Immunology, Department of Pathology, Faculty of Medicine and Health Sciences, Stellenbosch University, Cape Town, South Africa, 2 Division of Neonatology, Department of Paediatrics and Child Health, Faculty of Medicine and Health Sciences, Stellenbosch University, Cape Town, South Africa, 3 Division of Epidemiology and Biostatistics, Department of Global Health, Faculty of Medicine and Health Sciences, Stellenbosch University, Cape Town, South Africa, 4 Division of Paediatric Infectious Diseases, Department of Paediatrics and Child Health, Faculty of Medicine and Health Sciences, Stellenbosch University, Cape Town, South Africa

* kessendri@yahoo.com, kessendri.reddy@nhls.ac.za

## Abstract

### Background

Analysis of hospital-acquired bloodstream infection (HA-BSI) trends is important to monitor emerging antimicrobial resistance (AMR) threats and guide empiric antibiotic choices.

### Methods

A retrospective 10-year review of neonatal HA-BSI was performed at Tygerberg Hospital's neonatal unit in Cape Town, South Africa. Neonatal clinical and laboratory data from 2014 to 2018 (Period 2) was compared with published data from 2009 to 2013 (Period 1).

### Results

The neonatal unit's HA-BSI rate declined between periods from 3.9/1000 inpatient-days in Period 1 to 3.3/1000 inpatient-days in Period 2 (p = 0.002). Pathogen yield and blood culture contamination rate were unchanged (11.0% to 10.4%, p = 0.233; 5.1% to 5.3%, p = 0.636 respectively). Gram-negative pathogens predominated (1047/1636; 64.0%); *Klebsiella* species, *Staphylococcus aureus*, *Serratia marcescens*, *Enterococcus* species and *Acinetobacter baumannii* were the most frequent pathogens. Extended spectrum beta-lactamase production was observed in 319/432 (73.8%) of *Klebsiella* species, methicillin resistance in 171/246 (69.5%) of *Staphylococcus aureus* and extensive drug resistance in 115/137 (83.9%) of *Acinetobacter* species (2009–2018). The crude mortality rate of neonatal HA-BSI episodes increased from Period 1 to Period 2 from 139/717 (19.4%) to 179/718 (24.9%) (p = 0.014), but HA-BSI attributable mortality remained unchanged (97/139 [69.8%] vs 118/179

**Data Availability Statement:** All relevant data are within the manuscript.

**Funding:** No funding was specifically sought for the purposes of this study. This study was supported by National Health Laboratory Services Tygerberg. AD is supported by a US National Institutes of Health (NIH) Fogarty Emerging Global Leader Award K43 TW010682 and receives funding for neonatal infection research from the South African Medical Research Council. The funders had no role in study design, data collection and analysis, decision to publish, or preparation of the manuscript.

**Competing interests:** The authors have declared that no competing interests exist.

[65.9%], p = 0.542). The *in-vitro* activity of piperacillin-tazobactam and amikacin declined during Period 2 (74.6% to 61.4%; p<0.001).

## Conclusion

Although HA-BSI rates declined in the neonatal unit, antimicrobial resistance rates in BSI pathogens remained high. Continuous BSI surveillance is a valuable tool to detect changes in pathogen and AMR profiles and inform empiric antibiotic recommendations for neonatal units in resource-limited settings.

## Introduction

Infection is the third leading contributor to deaths in neonates (<28 days of life) globally. In 2018, an estimated 1 016 000 neonatal deaths occurred in sub-Saharan Africa, 23% of which were attributed to infection [1]. Hospitalised neonates in resource-limited settings are especially vulnerable to infection [2, 3] due to overcrowding, resource constraints and suboptimal infection prevention practices, among other factors [4]. In low-to-middle income countries (LMIC), the overall burden of neonatal infection is 3–20 fold higher than in high-income country settings [5] with reported rates of healthcare-associated infection in LMIC neonatal intensive care units (ICUs) ranging from 15–62 infections per 1000 patient-days [6]. Hospital-acquired bloodstream infections (HA-BSI) are the most frequent complication experienced, and disproportionately affect preterm neonates [3].

Data on the burden, spectrum and impact of BSI in hospitalised neonates in LMIC is scanty, owing to challenges in the diagnosis of neonatal sepsis [7], inequitable access to microbiology laboratory services, limited surveillance capability and inadequate reporting [3, 5]. Active surveillance of neonatal BSI rates is important for local benchmarking, regional and global comparisons of neonatal infection rates or density [5], detection of trends in antimicrobial resistance (AMR) and review of empiric antibiotic recommendations. Although standardised guidance exists for the first-line empiric therapy of suspected neonatal sepsis [8], there is considerable heterogeneity in second-line empiric regimens for HA-sepsis, which require knowledge of pathogen and AMR profiles in individual units [9]. Hospital datasets (with microbiological and clinical outcome data) are infrequently available to inform empiric antimicrobial recommendations at African country neonatal units.

Neonatal BSI surveillance also provides critical information on targeted interventions needed to prevent neonatal infections and deaths [10]. Previously published data from 2009–2013 in our unit [11] reported a neonatal HA-BSI rate of 3.9 per 1000 inpatient-days, with a predominance of Gram-negative pathogens (65.2%) and a crude mortality rate of 15.6%. Among neonatal HA-BSI episodes, AMR rates were substantial; 73.2% of *Klebsiella pneumoniae* isolates produced extended spectrum beta-lactamases, 89.9% of *Acinetobacter baumannii* isolates were extensively drug resistant and 66.1% of *Staphylococcus aureus* isolates were methicillin-resistant.

Given rising global AMR rates and the United Nations Sustainable Development Goal aiming to end preventable neonatal deaths [10], we described neonatal HA-BSI trends over 10 years at a tertiary neonatal unit in Cape Town, South Africa. We compared HA-BSI, AMR and BSI-attributable mortality rates between 2009–2013 and 2014–2018, and evaluated the antibacterial activity of piperacillin-tazobactam and amikacin as second-line empiric therapy for neonatal HA-sepsis from 2014–2018.

## Materials and methods

### Study design

We conducted a retrospective review of neonatal HA-BSI episodes at Tygerberg Hospital, Cape Town, South Africa, between 1 January 2014 and 31 December 2018 (Period 2) and compared the data with a previously published dataset from 1 January 2009 to 31 December 2013 (Period 1), using the same methodology [11].

### Study setting

Tygerberg Hospital is a 1384-bedded tertiary hospital serving the Cape Metro region's Northern and Eastern subdistricts and surrounding rural districts' healthcare facilities. An on-site tertiary obstetric unit manages approximately 8000 high-risk deliveries per annum, with a 37% low birth weight rate. In 2012, the neonatal unit expanded from 98 to 124 beds including a 12-bed medical/surgical neonatal intensive care unit, 2 high-dependency wards, 1 low-care ward and 1 kangaroo mother care ward. Common indications for neonatal admission are prematurity, perinatal asphyxia and neonatal sepsis. The unit is supported by an on-site diagnostic Microbiology laboratory, neonatologists, Paediatric Infectious Disease specialists and a dedicated hospital "Unit for Infection Prevention and Control" (UIPC). Continuous neonatal BSI surveillance is conducted by the UIPC nurse practitioner allocated to the neonatal, paediatric and obstetric service, with feedback to clinicians using a weekly neonatal HA-BSI line list and monthly BSI and blood culture contamination run-charts. A formal antimicrobial stewardship programme and a central line-associated bloodstream infection prevention programme were established in the neonatal wards in 2012.

### Investigation of suspected neonatal HA-BSI

A single blood culture sample is aseptically collected from neonates with clinical deterioration or signs of possible neonatal sepsis e.g. temperature or glycaemic instability, tachypnoea, tachycardia etc., at the discretion of the attending clinician. The automated BacT/Alert blood culture incubation system and BacT/Alert PF Plus bottles (Biomérieux, Marcy l'Ètoile, France) have been in use since April 2011; prior to this, the BACTEC system (Becton Dickinson, New Jersey, USA) and BACTEC Peds Plus/F bottles were used. For positive blood cultures, a Gram stain is made from the blood culture broth and the broth is subcultured and incubated overnight. Clinicians are telephonically alerted to potential pathogens on Gram stain during the laboratory's operational hours. Identification and susceptibility testing usually involves use of the automated VITEK® 2 system (Biomérieux, Marcy l'Ètoile, France) and/or disk diffusion testing, which are interpreted using annually published Clinical Laboratory Standards Institute (CLSI) breakpoints. Categorisation of likely resistance mechanisms was based on the data generated by the above routine phenotypic susceptibility testing; molecular investigation of these mechanisms and more specialised agar-based testing (such as to interrogate mechanisms of resistance to 3rd generation cephalosporins in chromosomal *ampC* beta-lactamase producers) are not routinely performed in our diagnostic laboratory.

### Management of suspected neonatal HA-BSI

Prior to mid-2016, neonates with suspected HA-BSI received meropenem while awaiting culture results, with addition of vancomycin in patients with risk factors for methicillin-resistant *Staphylococcus aureus* (MRSA). From mid-2016, the hospital's empiric antibiotic recommendation for HA-BSI was changed from meropenem to piperacillin-tazobactam and amikacin for clinically stable neonates awaiting culture results. This was based on a hospital-wide review

of culture and susceptibility results of blood cultures for patients (adult and paediatric) from all departments. The review was prompted by the emergence of carbapenem resistant Enterobacterales (CRE) in 2015 in the hospital, and the combination of piperacillin-tazobactam and amikacin was considered as part of the approach to reduce selective pressure exerted by the widespread administration of carbapenems for all cases of suspected HA infection. For critically ill neonates with bacteraemia, septic shock or if meningitis could not be excluded, high-dose meropenem with or without vancomycin remained the empiric therapy of choice for HA-BSI. In neonates, the threshold to revert to meropenem with or without vancomycin remains low; this is decided on a case-by-case basis by the neonatal consultant in consultation with Microbiology specialists/trainees and/or Paediatric Infectious Disease specialists. Empiric therapy is de-escalated or targeted based on culture results, during daily evaluations by neonatal consultants. Antifungal prophylaxis with fluconazole is not used in the neonatal unit.

## Data collection

Any infant from whom a blood culture was submitted beyond day 3 of hospitalisation in a neonatal ward was included in the HA-BSI data extraction from the National Health Laboratory Service database. Data on neonatal outcome (death, discharge or transfer) was obtained from the electronic hospital admission system. No notable changes to laboratory processing occurred between 2009 and 2018, aside from a shift in the automated blood culture system used in 2011. Notable changes to the CLSI breakpoints for cefotaxime/ceftriaxone, ceftazidime and cefepime, and the carbapenems occurred in 2010, with lower breakpoint values used to define susceptibility and resistance. For ertapenem, these values were revised slightly upwards again in 2012 [12].

## Study definitions

Organisms were classified using the United States Centre for Disease Control list of pathogens and contaminants [13]. HA-BSI was defined as a positive pathogen-containing blood culture in any neonate hospitalised for 3 or more days (but within 100 days) in any neonatal ward. We used a pragmatic threshold of infants up to 100 days of postnatal age, as prolonged hospital stay in preterm infants is a recognised risk factor for HA infection; infants developing HA infection after 100 days were no longer considered part of the neonatal population. Any blood culture isolating the same pathogen within 14 days of the original culture, was considered to represent a single BSI episode. Polymicrobial blood cultures with pathogens and contaminants were classified as blood cultures containing pathogens. Isolation of two or more coagulase-negative staphylococci (CoNS) with the same species designation and/or antibiogram, from two separate blood culture draws collected within 24 hours, were considered as pathogens [11]. Resistance mechanisms were categorised based on phenotypic results: members of the Enterobacterales group were classified as harbouring extended-spectrum beta-lactamases (ESBL), exhibiting carbapenem resistance (CRE) or being 3rd-generation cephalosporin resistant (3GCR). This latter category was used for chromosomal *ampC* beta-lactamase producing organisms resistant to 3rd generation cephalosporins, where clear differentiation between an ESBL enzyme and *ampC* beta-lactamase hyperproduction could not be made using susceptibility testing on VITEK® 2 or disk diffusion testing. Non-fermenting Gram-negatives were classified using standard definitions [14]. The crude mortality rate was calculated as all deaths among neonates with BSI episodes, as a proportion of all neonates with BSI episodes. BSI-attributable mortality was defined as death within 72 hours of blood culture collection [15], as a proportion of all deaths among neonates with BSI episodes. Assessment of the *in-vitro* activity of antimicrobials was based on the results reported on the laboratory information system

with no additional interpretation for the Gram-negative pathogens. For the Gram-positive pathogens, limited inferences were made at the discretion of a Microbiology specialist for other beta-lactam agents based on penicillin/ampicillin/cloxacillin activity, as not all beta-lactams were tested directly. Antibiotics reported as intermediate were classified as not having *in-vitro* activity. The proportion of antibacterial activity for HA-BSI (concordance of pathogens' AMR profiles and *in-vitro* spectrum of action of the empiric regimen) was calculated for Period 2 (2014–2018) in view of the change in empiric neonatal HA-BSI treatment from meropenem and/or vancomycin to piperacillin-tazobactam and amikacin in mid-2016.

### Statistical analysis and ethics approval

Statistical analysis was performed using Stata v15.1 (StataCorp, College Station, Texas) and Epicalc 2000 v1.02 (Brixton Health, United Kingdom), using an α-level of 0.05 with a corresponding 95% confidence interval (CI), for descriptive statistics and hypothesis testing. Two-sided p-values were consistently used. For normally-distributed continuous variables, means and standard deviations were calculated. Medians and interquartile ranges (IQR) were used for non-normally distributed continuous variables. Independent t-tests were used to compare continuous variables with normal distributions, with the non-parametric alternative of the Mann-Whitney U-test where the variables were not normally distributed. Binary categorical variables were analysed using the χ2 test or Fisher's exact test. Trend in proportions per year was analysed using a χ2 test for linear trend. Poisson regression analysis was used to assess trends in HA-BSI rates over time. A waiver of individual informed consent was granted through the Human Health Research Ethics Committee of Stellenbosch University (S18/10/262) and approval was granted by Tygerberg Hospital management.

### Results

The overall HA-BSI rate (2009–2018) was 3.6 per 1000 inpatient-days (95% CI 3.4–3.8), or 27.5 per 1000 admissions (95% CI 26.1–28.9) (Table 1). Although the HA-BSI rate declined between Period 1 and 2 from 3.9 to 3.3 per 1000 inpatient-days (p = 0.002), the HA-BSI trend did not decline significantly over the 10-year study period (Fig 1). Pathogen yield and blood culture contamination rates were unchanged between time periods (Table 1, Fig 1).

In both periods, Gram-negative pathogens predominated; *Klebsiella* species, *Staphylococcus aureus*, *Serratia marcescens*, *Enterococcus* species and *Acinetobacter baumannii* were the most frequent pathogens overall (Table 1). Between Period 1 and Period 2, there were significant increases in *S. marcescens* (16.2% to 22.0%, p = 0.021) and *Enterobacter* species (2.9% to 5.9%, p = 0.028) as a proportion of Gram-negative infections, and a significant reduction in the proportion of Gram-negative BSI caused by *Klebsiella* species (45.3% to 37.3%, p = 0.011).

Among Gram-positive pathogens, there was a substantial reduction in the proportion of Group B streptococcus between the two periods (14.8% to 7.6%, p = 0.013). Methicillin-resistant *S. aureus* was also a major contributor to HA-BSI, with no change in the AMR profile between study periods. Yeasts were responsible for 4.1% of HA-BSI overall and remained stable between the two time periods (Table 1). Coagulase-negative staphylococci were considered pathogens in 5 neonates in Period 1 and in 2 neonates in Period 2, accounting for only 0.5% of the blood cultures with pathogens (7/1435, 95% CI 0.2–1.1%).

For the top Gram-negative pathogens over the 10-year period, 73.8% of *Klebsiella* species were noted to harbour ESBLs phenotypically (95% CI 69.4–77.9%), 13.0% of *S. marcescens* isolates were 3GCR (95% CI 8.8–18.7%), and 83.9% of *A. baumannii* isolates were extensively drug resistant (95% CI 76.5–89.5%); the methicillin-resistant phenotype predominated in *S. aureus* (69.5%, 95% CI 63.3–75.1%). The only statistically significant difference in resistance

**Table 1. Comparison of key variables in neonatal hospital-acquired bloodstream infection at Tygerberg Hospital from 2009–2018, and comparison between the Period 1 (2009–2013) and Period 2 (2014–2018).**

| Variable | Overall estimate (2009–2018) Number (%) [95% CI] | Period 1 (2009–2013) Number (%)* [95% CI] | Period 2 (2014–2018) Number (%)* [95% CI] | p-value (two-sided) |
|---|---|---|---|---|
| Number of admissions | 52204 | 23920 | 28284 | - |
| Number of inpatient-days | 400614 | 183846 | 216768 | - |
| Blood culture collection rate per total admission count | 13461/52204 (25.8) [25.4–26.2] | 6521/23920 (27.3) [26.7–27.8] | 6940/28284 (24.5) [24.0–25.0] | <0.001 |
| HA-BSI rate/1000 inpatient-days[a] | 3.6 (3.4–3.8) | 3.9 (3.6–4.2) | 3.3 (3.1–3.6) | 0.002 |
| HA-BSI rate/1000 admissions[a] | 27.5 (26.1–28.9) | 30.0 (27.9–32.2) | 25.4 (23.6–27.3) | 0.002 |
| Absolute blood culture positivity rate, including duplicate blood cultures[b] | 2544/13461 (18.9) [18.2–19.6] | 1145/6521 (17.6) [16.7–18.5] | 1399/6940 (20.2) [19.2–21.1] | <0.001 |
| Pathogen yield (deduplicated)[b] | 1435/13461 (10.7) [10.2–11.2] | 717/6521 (11.0) [10.3–11.8] | 718/6940 (10.4) [9.6–11.1] | 0.233 |
| Blood culture contamination rate (deduplicated)[b] | 701/13461 (5.2) [4.8–5.6] | 333/6521 (5.1) [4.6–5.7] | 368/6940 (5.3) [4.8–5.9] | 0.636 |
| Number of BSI episodes | 1435 | 717 | 718 | - |
| Monomicrobial BSI episodes | 1225/1435 (85.4) [83.4–87.1] | 650/717 (90.7) | 575/718 (80.1) | <0.001 |
| Proportion pathogen counts[c] | | | | |
| Gram-negatives | 1047/1641 (63.8) [61.4–66.1] | 519/796 (65.2) [61.7–68.5] | 528/845 (62.5) [59.1–65.7] | 0.275 |
| Gram-positives | 522/1641 (31.8) [29.6–34.1] | 244/796 (30.7) [27.5–34.0] | 278/845 (32.9) [29.8–36.2] | 0.356 |
| Yeasts | 67/1641 (4.1) [3.2–5.2] | 33/796 (4.2) [2.9–5.8] | 34/845 (4.0) [2.8–5.6] | >0.999 |
| Pathogen proportion for the 10 most frequently isolated pathogens overall, by organism grouping (rank) | | | | |
| *Klebsiella* species (1) | 432/1047 (41.3) [38.3–44.3] | 235/519 (45.3) [40.9–49.7] | 197/528 (37.3) [33.2–41.6] | 0.011 |
| *Staphylococcus aureus* (2) | 246/522 (47.1) [42.8–51.5] | 112/244 (45.9) [40.0–52.4] | 134/278 (48.2) [42.2–54.2] | 0.662 |
| *Serratia marcescens* (3) | 200/1047 (19.1) [16.8–21.6] | 84/519 (16.2) [13.2–19.7] | 116/528 (22.0) [18.6–25.8] | 0.021 |
| *Enterococcus* species (4) | 198/522 (37.9) [33.8–42.3] | 88/244 (36.1) [30.1–42.5] | 110/278 (39.6) [33.8–45.6] | 0.464 |
| *Acinetobacter baumannii* (5) | 137/1047 (13.1) [11.1–15.3] | 69/519 (13.3) [10.6–16.6] | 68/528 (12.9) [10.2–16.1] | 0.914 |
| *Escherichia coli* (6) | 119/1047 (11.4) [9.5–13.5] | 58/519 (11.2) [8.7–14.3] | 61/528 (11.6) [9.0–14.7] | 0.924 |
| Group B streptococcus (8) | 57/522 (10.9) [8.4–14.0] | 36/244 (14.8) [10.7–19.9] | 21/278 (7.6) [4.9–11.5] | 0.013 |
| *Enterobacter* species (7) | 46/1047 (4.4) [3.3–5.9] | 15/519 (2.9) [1.7–4.8] | 31/528 (5.9) [4.1–8.3] | 0.028 |
| *Candida albicans* (9) | 33/67 (49.3) [37.0–61.6] | 18/33 (54.6) [36.6–71.5] | 15/34 (44.1) [27.6–61.9] | 0.542 |
| *Pseudomonas aeruginosa* (10) | 32/1047 (3.1) [2.1–4.3] | 12/519 (2.3) [1.3–4.1] | 20/528 (3.8) [2.4–5.9] | 0.227 |
| Crude mortality[d] | 318/1435 (22.2) [20.1–24.4] | 139/717 (19.4) [16.6–22.5] | 179/718 (24.9) [21.8–28.3] | 0.014 |
| BSI-attributable mortality[e] | 215/318 (67.6) [62.1–72.7] | 97/139 (69.8) [61.3–77.1] | 118/179 (65.9%) [58.4–72.7] | 0.542 |
| Crude mortality by pathogen group[f] | | | | |
| Gram-negative | 153/806 (19.0) [16.4–21.9] | 78/433 (18.0) [14.6–22.0] | 75/373 (20.1) [16.2–24.6] | 0.506 |
| Gram-positive | 27/358 (7.5) [5.1–10.9] | 15/186 (8.1) [4.7–13.2] | 12/172 (7.0) [3.8–12.2] | 0.850 |
| Yeast | 6/60 (10.0) [4.1–21.2] | 3/31 (9.7) [2.5–26.9] | 3/29 (10.3) [2.7–28.5] | 0.731 |

CI: confidence interval; HA-BSI: hospital-acquired bloodstream infection; MRSA: methicillin-resistant *S. aureus*; MDR: multidrug-resistant; BSI: bloodstream infection.

[a]Number (%) [95% CI] applies to all variables listed with the exception of HA-BSI rate, expressed per 1000 inpatient-days, or per 1000 admissions.

[b]Absolute blood culture positivity rate comprises all neonatal blood cultures flagging positive on or after day 3 of admission, including duplicate blood cultures, over all neonatal blood cultures submitted, while pathogen yield and contamination rate refer to deduplicated blood cultures as a proportion of all neonatal blood cultures submitted.

[c]An additional 5 pathogens were unclassified in Period 2 (2014–2018), including *Rhodotorula* species, *Wickerhamomyces anomalus*.

[d]Crude mortality defined as deaths in neonates with pathogen-containing BSI, as a proportion of all neonates with BSI episodes.

[e]BSI-attributable mortality defined as deaths occurring within 72 hours of blood culture collection, as a proportion of all deaths in neonates with BSI episodes.

[f]Monomicrobial cultures only.

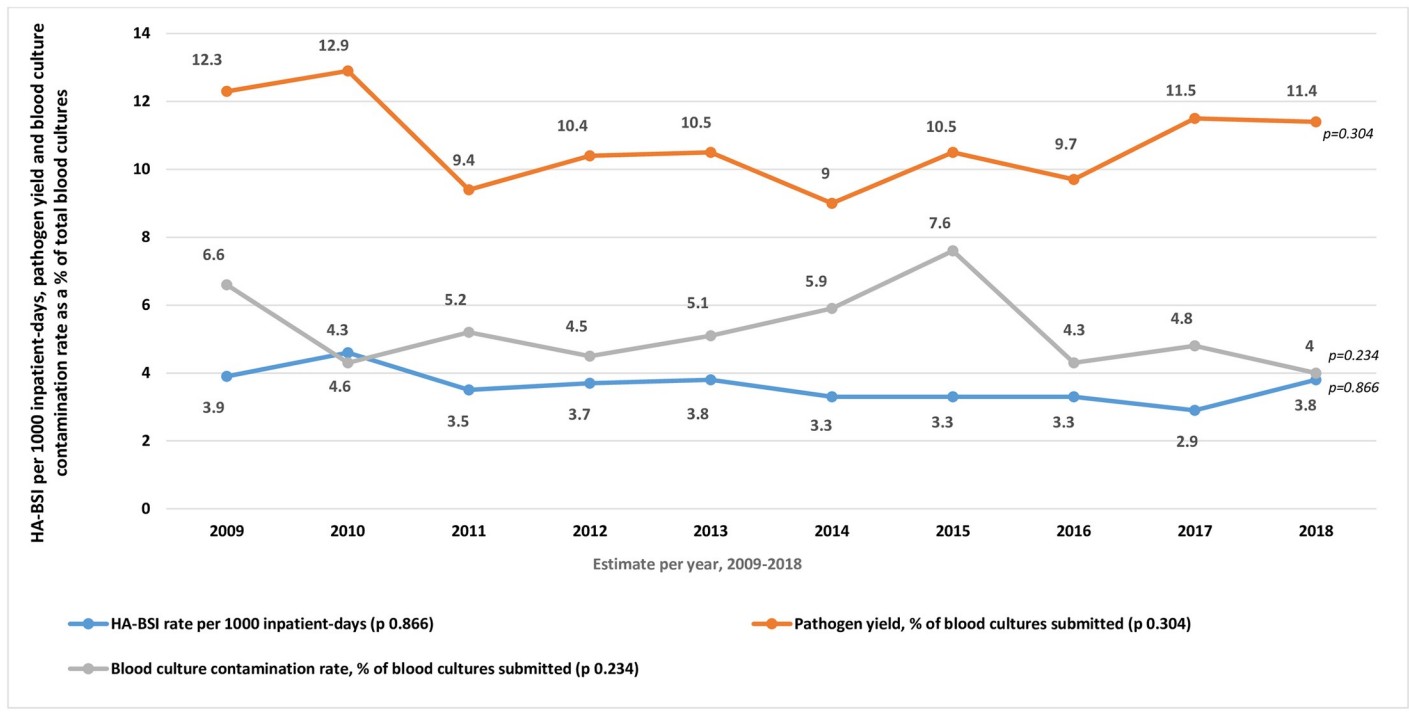

**Fig 1. Trends in neonatal hospital-acquired bloodstream infection (HA-BSI) rate, pathogen yield and blood culture contamination rate (2009–2018).** *HA-BSI rate calculated using number of HA-BSI episodes over total inpatient-days multiplied by 1000, from 2009–2018. HA-BSI comparisons made using Poisson regression. Pathogen yield and blood culture contamination rate calculated as number of pathogen-containing and contaminant-containing blood cultures respectively, as a proportion of all blood cultures submitted from 2009–2018; compared using a $\chi^2$ test for linear trend. Two-sided p-values were used.

profile among the most frequently isolated pathogens between Period 1 and Period 2, was observed for *S. marcescens* (Fig 2). The AMR profiles of frequently isolated pathogens are summarised in Fig 2. Notable increases in a more resistant phenotype were noted for *S. marcescens* in 2015 and 2016 (no corresponding documented outbreak), methicillin-resistant *S. aureus* in 2017 (no corresponding documented outbreak) and extensively drug resistant *A. baumannii* in 2017 (corresponding to a documented outbreak) [16]. No CREs were isolated from 2009–2013, in contrast to the 4 CREs isolated in Period 2, all of which occurred between 2017 and 2018. These were exclusively found in *K. pneumoniae*, and *bla*NDM-1 enzymes were detected in all 4 isolates at the reference laboratory at the National Institute of Communicable Diseases.

Although not exclusively within the study period, a CRE outbreak comprising mostly *K. pneumoniae* strains was detected on the neonatal platform from late December 2018 and continued into 2019. Molecular investigation of this outbreak confirmed carbapenemases belonging to both the *bla*OXA-48 and variants and *bla*NDM-1 families. Rep-PCR analysis of 21 *K. pneumoniae* isolates revealed that the majority of these isolates were not genetically related (3 *K. pneumoniae* clusters consisting of 2 isolates each, others not related) (personal communication, Kedisaletse Moloto).

The crude mortality rate was substantially higher in Period 2 compared with Period 1 (24.9% vs 19.4%, p = 0.014), although BSI-attributable mortality remained unchanged (69.8% vs 65.9%, p = 0.542) (Table 1). Crude mortality by pathogen group was stable over the two time periods for neonates with monomicrobial BSI, with Gram-negative BSI associated most strongly with crude mortality overall (19.0% vs 7.5% vs 10.0% for Gram-negative, Gram-positive and yeasts respectively, p<0.001).

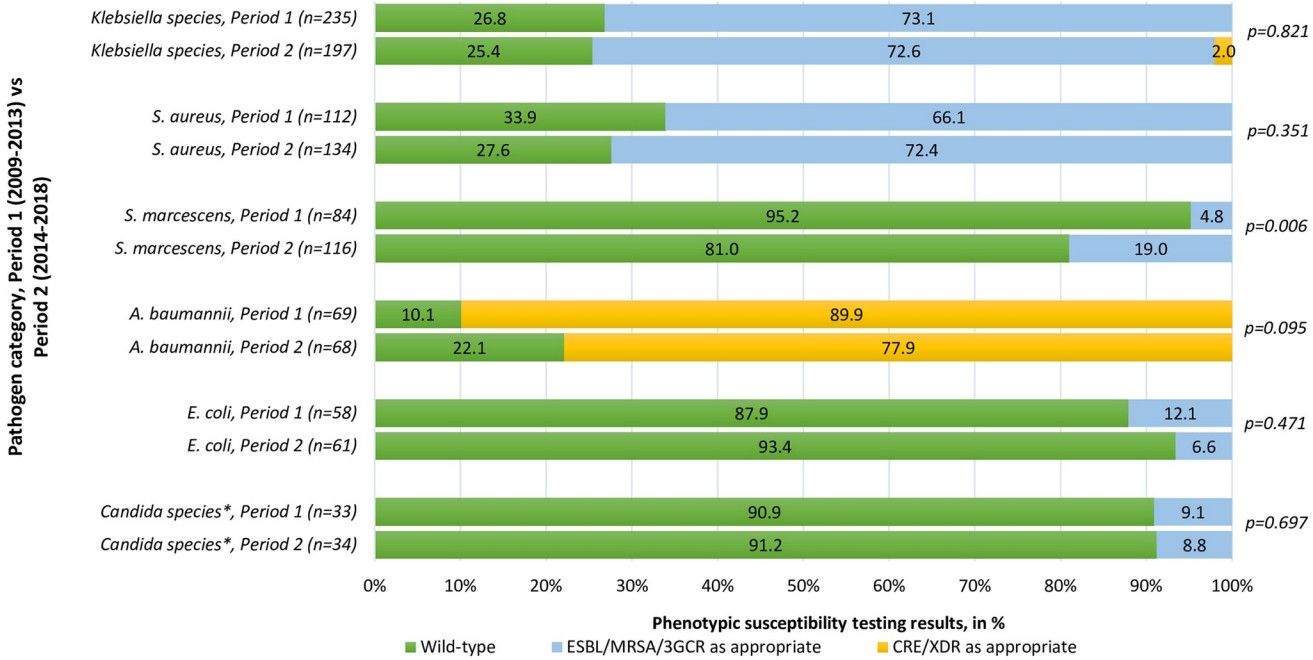

**Fig 2. Phenotypic susceptibility patterns for the most commonly isolated pathogens in neonatal hospital-acquired bloodstream infections, Period 1 (2009–2013) vs Period 2 (2014–2018).** *Phenotypic susceptibility patterns established using the relevant year's Clinical and Laboratory Standards Institute (CLSI) breakpoints. Isolates classified according to standard definitions for wild-type and acquired resistance, and reported as a proportion of the total number of isolates per species. Two-sided p-values were used. ESBL: extended-spectrum β-lactamase, detected by third-generation cephalosporin resistance; MRSA: methicillin-resistant *S. aureus*, detected by cefoxitin resistance;3GCR: 3rd-generation cephalosporin resistance in chromosomal *ampC* beta-lactamase producers; CRE: carbapenem-resistant Enterobacterales, detected by reduced susceptibility to any carbapenem (usually ertapenem), in isolates not known to have chromosomally-mediated mechanisms for reduced susceptibility, XDR: extensively drug resistant, classified using a standard definition [14]. *Including *Candida* spp. with established fluconazole resistance, *C. krusei* and *C. glabrata*, which were the only isolates in which fluconazole resistance was found. No *C. auris* isolates were detected in this setting during the period of study.

From 2014 to 2018, we reviewed 682 individual neonatal HA-BSI episodes for which piperacillin-tazobactam and/or amikacin susceptibility was reported (Gram-negatives) or could be inferred (Gram-positives, based on penicillin/ampicillin/cloxacillin susceptibility). The regimen of piperacillin-tazobactam and amikacin displayed *in-vitro* activity for 437 episodes overall (64.1%, 95% CI 60.3–67.7%), although a reduction in *in-vitro* antibacterial activity was observed from 74.6% in 2014 to 61.4% in 2018 ($\chi^2$ test for linear trend p<0.001).

## Discussion

In the second half of the decade studied, neonatal admissions increased by an average of 870 per year owing to changes in defined regional populations serviced by referral hospitals, as well as increasing urbanisation and population growth in the Western Cape province [17]. Despite a higher number of admissions within Period 2, there were fewer blood cultures submitted, suggesting lower rates of suspected infection in these infants. This is supported by the significant reduction in HA-BSI rate between Periods 1 and 2, which is encouraging in the context of increased admissions and neonatal unit occupancy rates. This finding may reflect normal variation over longer time periods, may reflect the impact of a more established antimicrobial stewardship programme including exposure to Microbiology specialists, infection control practitioners and Infectious Disease specialists, or may be indicative of the heightened emphasis on antimicrobial stewardship, including diagnosing infection, that is becoming a

feature of medical training in South Africa [18]. It is challenging to draw conclusions from this finding, however, as the 5-year threshold used is arbitrary; the year-on-year analysis assessing trends over the 10-year period (Fig 1) is likely to be a more robust assessment of this variable and showed no difference in HA-BSI over the full study period.

Our neonatal unit's HA-BSI rate remains much lower than that reported from similar settings (11.6 HA-BSI per 1000 inpatient-days in Cameroon [19]; 17.0 HA-BSI per 1000 inpatient-days in India [20]; 13.6 culture-proven hospital-acquired infections per 1000 inpatient-days in Sri Lanka [21]; 35.6 late-onset BSI per 1000 admissions in Nigeria [22]; 283.9 HA-BSI per 1000 admissions in a neonatal ICU in Egypt [23]; 52.4 late-onset BSI per 1000 admissions in a neonatal ICU in Malaysia [24]). The use of different denominators, and the reporting of different sepsis types, settings and populations precludes more meaningful global comparison of these estimates [25, 26].

As previously noted in our centre, Gram-negative pathogens predominated in HA-BSI episodes, most notably *Klebsiella* species, *S. marcescens*, *A. baumannii* and *E. coli*. This pattern has also been reported from other LMIC countries [3, 27], and is in contrast to HA-BSI episodes in high-income settings where Gram-positive pathogens predominate [28–31]. The contrast in causative pathogen types for HA-BSI is potentially due to differences in climate, clinical practice, quality of hospital water sources, sanitation and hygiene status, infection control and antibiotic prescribing practices.

*S. marcescens* remains an important pathogen in our neonatal unit and is associated with mortality and severe neurological impairment [32, 33], a finding which has not yet been described in other LMIC settings [25, 32, 34–39]. *A. baumannii* is responsible for a large proportion of neonatal infections and mortality, and caused a major outbreak of carbapenem-resistant *A. baumannii* in our unit in 2017 [16]. This organism is substantially more prevalent in South Africa [40] than has been reported elsewhere on the continent [3, 22, 23], although it features more prominently in Egypt [23], Ghana [36], Tanzania [41] and South Asia [20, 27].

As expected, *S. aureus* and *Enterococcus* species were responsible for most HA-BSI episodes due to Gram-positive pathogens, and these remain associated with significantly lower mortality than Gram-negative and fungal BSI pathogens [11]. Pathogen spectrum varies geographically, with most centres reporting a predominance of coagulase-negative staphylococci [25, 26, 31, 39, 42, 43]. Coagulase-negative staphylococci were defined as pathogens in this context using a strict definition, likely resulting in underestimation of their contribution to overall HA-BSI in this study. The reduction in Group B streptococcus prevalence in the context of neonatal HA-BSI is challenging to interpret but may be due to circulating strains of differing virulence [44].

The majority of HA-BSI episodes caused by three of the major pathogens (*Klebsiella* species, *S. aureus* and *A. baumannii*) were primarily associated with antibiotic resistant phenotypes. The AMR rates documented in this neonatal unit exceed that reported from other regions [3, 28, 35] and have remained elevated over both time periods, suggesting significant sustained antibiotic selection pressure or persistence in the hospital environment. Potentially, this finding may also represent the presence of circulating strains that more readily develop and harbour resistance. Lower breakpoints for extended-spectrum cephalosporins in the Enterobacterales group were introduced by CLSI in 2010 [12] and were implemented in our laboratory that year; this may have resulted in underestimation of ESBL-producing isolates in 2009 compared with the rest of the study period (2010–2018). The majority of the data, however, was analysed using the same cephalosporin breakpoints. Carbapenem-resistant Enterobacterales did not feature prominently during the majority of this study period, but emerged in 2017 and 2018 and may have been the forerunners of an outbreak on the neonatal platform which

began in late December 2018 and continued into 2019, accounting for 25% of *K. pneumoniae* BSI in 2019 (personal communication A Dramowski).

Fungal pathogens remain an uncommon cause of neonatal HA-BSI in our setting, with a prevalence of about 4%, similar to that reported in Australia, North America and China [30, 31, 39], but in contrast to the higher numbers observed elsewhere [29, 35, 45]. There was a trend towards a reduction in *Candida albicans* and an increase in *C. parapsilosis* and *C. krusei* in Period 2. Fluconazole-resistant *C. parapsilosis* in the setting of neonatal infection is well-documented in South Africa [46], but has been uncommon in this neonatal unit to date. Continued monitoring with specific vigilance to detect emergence of *C. auris* is needed, as the pathogen has been documented in neonatal units in the Gauteng province of South Africa [47].

The crude HA-BSI mortality rate increased significantly between the study periods (19.4% to 24.9%), while BSI-attributable mortality remained stable. Global estimates of healthcare-associated infection (including BSI) mortality range between 10–20% [2, 35, 38, 42, 45, 48, 49]. There are no neonatal BSI data that report BSI-attributable mortality in Africa. The increased crude mortality in Period 2 may reflect improved access to medical care for extremely preterm neonates, resulting in smaller babies being exposed to interventions and surviving for more than 72 hours, thereby increasing their risk of HA infection.

When comparing the pathogen profiles between the two time periods, reassuring reductions were seen in the prevalence of *Klebsiella* species. *Klebsiella* species are consistently reported as one of the main Gram-negative contributors to HA infection in neonates [3, 27, 35, 36, 38, 41, 43, 50] and are adept at accumulating transferable resistance mechanisms such as ESBLs and carbapenemases [51]. The resistance rates for this pathogen remained stable over time, but carbapenem-resistant *K. pneumoniae* became a concern in the last month of the study period. For *S. marcescens* however, both the absolute number and prevalence of the resistant phenotype increased significantly in Period 2. This increase can be at least partially explained by undetected *S. marcescens* clusters, but vigilance should be maintained due to its virulent nature, its intrinsic resistance to colistin and the emergence of transferable carbapenemases on the neonatal platform.

We report a decline in the *in-vitro* activity of piperacillin-tazobactam and amikacin between 2014 and 2018 in this cohort. Prior to 2016, meropenem had been the mainstay of empiric therapy for neonatal HA-BSI. Piperacillin-tazobactam plus amikacin was re-introduced in mid-2016 as the local empiric antibiotic recommendation for suspected HA-infection in stable neonates without meningitis, in an effort to limit carbapenem use and selective pressure. This remains of critical importance in view of the recent rise in CRE and the high prevalence of extensively drug resistant *A. baumannii* in our unit. The combination of piperacillin-tazobactam and amikacin is still effective *in-vitro* against the majority of isolates in the latter 5 years of this study, but this reduction in activity is of concern and should be evaluated in more depth, specifically with reference to the absolute numbers of neonates (culture-positive and culture-negative) started on this regimen and their clinical outcomes. This observation is limited by the fact that *in-vitro* susceptibility does not necessarily equate to clinical response, not all neonates with suspected HA sepsis have positive blood cultures, and not all neonates with suspected HA-infection have a cultivatable bacterial cause of sepsis. The selective pressure exerted by the empiric use of this regimen in suspected HA sepsis, or the occurrence of outbreaks, may be contributing factors in the declining antibiotic activity rate of piperacillin-tazobactam plus amikacin.

Limitations of this retrospective study include the use of record review, although the large sample size and combination of independently curated databases mitigated the risks associated with missing information. Neonatal admission denominator data using the hospital

information system may have overestimated the actual number of neonates admitted, as movement between wards was counted as a new admission; however, the same methods were employed by the hospital information system throughout the decade under study. Practices in aseptic blood culture collection and inoculum volume may have varied over time. More detailed analysis of the isolates in terms of molecular mechanisms of resistance and strain relatedness, particularly for common organisms in our setting such as *S. marcescens*, could not be performed due to the retrospective nature of the study. Lowering of the CLSI breakpoints for many of the beta-lactam antibiotic agents for the Enterobacterales group occurred in 2010 (with an additional amendment in 2012 for ertapenem); this may have influenced comparability of ESBL rates in 2009 with the rest of the data. This is likely to be of minimal significance as data from 2010–2018, the bulk of the study period, was analysed using the same breakpoint values, and CRE isolates only emerged in 2017/2018. The lack of comprehensive clinical information precluded thorough assessment of contaminants that may have been associated with infection (such as coagulase-negative staphylococci). This study did not take into account non-BSI related factors that influence neonatal outcome, such as a decision to withdraw care based on a poor prognosis or underlying congenital abnormalities. Lastly, the occurrence of pathogen clusters or outbreaks may have impacted on the pathogen and AMR profiles noted.

## Conclusions

The rates of HA-BSI declined between periods but blood culture contamination and pathogen yield remained stable between 2009 and 2018, despite increasing admission volumes on a platform with constrained physical and human resources. Gram-negative pathogens remain the predominant cause of HA-BSI, with substantial antibiotic resistance and high crude mortality rates. The activity of piperacillin-tazobactam and amikacin declined from 2014 to 2018, potentially due to its use in empiric treatment of HA infection. Ongoing neonatal HA-BSI surveillance and timely feedback to clinicians, in combination with review of empiric recommendations and targeted infection prevention interventions, are essential activities in LMIC neonatal units.

## Acknowledgments

The authors thank the patients, caregivers and staff of the Tygerberg Hospital Neonatal unit, as well as National Health Laboratory Services Tygerberg and Mr. Darryl Visagie and his team at Tygerberg Hospital's information system (for admission dates, mortality) and Mr. Hewitt de Jager for data on total inpatient days. We thank Dr. Mené van der Westhuyzen who assisted in extracting data on gestational age and birthweight of neonates who demised in Period 2. We also thank Mr. Wessel Kleynhans for facilitating access to data from 2014 and 2015 from the laboratory information system, and Mr. Jeremy Goodway for his technical assistance.

## Author Contributions

**Conceptualization:** Adrie Bekker, Angela Dramowski.

**Data curation:** Kessendri Reddy, Adrie Bekker, Angela Dramowski.

**Formal analysis:** Kessendri Reddy, Tonya M. Esterhuizen.

**Investigation:** Kessendri Reddy, Andrew C. Whitelaw.

**Methodology:** Kessendri Reddy, Tonya M. Esterhuizen, Angela Dramowski.

**Project administration:** Kessendri Reddy.

**Resources:** Kessendri Reddy, Adrie Bekker, Andrew C. Whitelaw, Angela Dramowski.

**Supervision:** Angela Dramowski.

**Validation:** Kessendri Reddy, Adrie Bekker, Tonya M. Esterhuizen, Angela Dramowski.

**Visualization:** Kessendri Reddy, Tonya M. Esterhuizen, Angela Dramowski.

**Writing – original draft:** Kessendri Reddy.

**Writing – review & editing:** Kessendri Reddy, Adrie Bekker, Andrew C. Whitelaw, Tonya M. Esterhuizen, Angela Dramowski.

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
