## [Decision Letter · Decision Letter 0]

9 Nov 2020

PONE-D-20-25473

A retrospective analysis of pathogen profile, antimicrobial resistance and mortality in neonatal hospital-acquired bloodstream infections from 2009-2018 at Tygerberg Hospital, South Africa

PLOS ONE

Dear Dr. Reddy

Thank you for submitting your manuscript to PLOS ONE. After careful consideration, we feel that it has merit but does not fully meet PLOS ONE’s publication criteria as it currently stands. Therefore, we invite you to submit a revised version of the manuscript that addresses the points raised during the review process.

A retrospective review of hospital- acquired bloodstream in infections at a large tertiary care neonatal unit in a Western Cape Province, South Africa was conducted. Over an overall 10-year period (two five-year periods), the authors note a reduction in HA-BSI. The authors note high rates of resistance during the study period, and a decline in the in vitro susceptibility of piperacillin tazobactam and amikacin. These findings are important and the manuscript is a valuable contrubution to the understanding of South African and global burden of neonatal sepsis.

Major comments:

The basis of the recommendation to use piperacillin tazobactam + amikacin in the latter half of the study requires further explanation. Data from the first half of the study period was published in 2015 and already demonstrated a 73% ESBL rate. Was this antibiogram-directed? If not, what was the basis of selecting these two agents for the unit?

The author does not differentiate between derepressed AmpC and ESBLs among S.marcescens isolates. Considering the large proportion of these isolates further detai, if available retrospectively, would be beneficial.

The resistance phenotypes are brushed over fairly superficially. More detail can be provided in the results – specifically, what proportion of isolates were XDR and PDR. Which carbapenemases enzymes were detected in the isolates? Was there an outbreak of K.pneumoniae CRE in 2019 to account for the 25% CRE among K.pneumoniae in that year?

A significant decrease in HA-BSI rates is noted in the latter half of the study period. However, these results are not discussed in the manuscript – no possible reasons for the decrease are suggested.

Elaborate on why a retrospective folder review was performed on patients who died. Was it to confirm that there was no other attributable cause of death?

More specific comments to address:

Line 83: Study design. A definition for an episode of HA-BSI should be included in the methods. Although alluded to in data collection line 124, this should be specifically stated and included in the study definitions section. Investigation of suspected neonatal HA-BSI: Specify if resistance mechanisms were confirmed using other methods e.g. to differentiate between ESBL and AmpC production among Serratia marcescens. Line 116: Management of suspected neonatal HA- BSI. Explain the basis of this recommendation – the paper referenced in this section of the manuscript by Dramowski in 2015 describes a 73% ESBL rateLine 126: Data collection. Please clarify if 100 days was chosen at random or if this was the maximum number of days at which HA-BSI was observed.

 Lne 209: in the Methods section, study definitions are provided including XDR and pandrug resistance. However, there is no comment on the proportion of isolates which were XDR or PDR in the results.

 Fiure 2: Considering the larger proportion of AmpC producing S.marcescens isolates presented, was it possible to determine which isolates were true ESBL’s and which were derepressed AmpC’s? If this distinction cannot be made due to the nature of data collection, then this should be stated in the methods section, and a limitation added to the discussion.

Carbapenem resistant Enterobacterales and Acinetobacter baumannii species are on the WHO Priority Pathogen’s Critical List. Additional details regarding these in the results and discussion would increase the impact of the manuscript.Figure 2: There appears to have been a change in epidemiology of CRE’s during period 1 and Period 2. It would be useful to elaborate, if possible, on the change. Were E.coli CRE’s due to the same carbapenemases enzyme as K.pneumoniae CRE’s in the latter part of the study?In the discussion, the authors mention a personal communication that 25% of K.pneumoniae isolates in 2019 were carbapenem resistant. However, this data is not reflected in this section of the results. Did this finding correspond to a documented outbreak in the unit?

In the discussion:A significant decrease in HA- BSI rate is reported, however the author does not offer possible reasons for this reductionLine 279: The comment about reduced virulence of circulating strains should be referencedLine 287: As per comment in results section. Over the 10- year study period, could changes in CLSI interpretive criteria have affected resistance reporting rates for ESBL’s and CRE’s? If so, these need to be discussed.

 Conclusion: In addition to surveillance, the value of unit antibiogram provision to review empiric recommendations should be included.

Minor Comments:

Line 37: Abstract Sentence amendment: " The crude mortality rate of neonatal HA-BSI increased from Period 1 to eriod 2 from 139/717 (19.4%) to 179/718 (24.9%)

Line 60: remove comma before "and inadequate reporting”, line 124, use " in a neonatal ward"  in place of "on", line 286, use " but have emerged in the neonatal unit" in place of "on"

Line 129: suggest reword for clarity: Retrospective folder review was conducted for patients who died to determine if the case would be categorized as attributable or crude mortality   

Line 242: Where the term “cover” is used to imply in vitro activity of an antibacterial, this should be changed to “antibacterial activity for empiric use”

Line 246: Discussion. Consider rewording “shift in draining patterns” to allow for improved understanding by an international readership – consider “changes to defined regional populations serviced by hospitals”

Line 246: Add “increased” urbanization

Line 248: There were “fewer” blood cultures submitted

References:

o     Please check references – The reference list appears twice

o     Journal title incorrectly captured by reference manager- reference 13

We look forward to receiving your revised manuscript.

Kind regards,

Adriano Gianmaria Duse, MD

Academic Editor

PLOS ONE

Journal Requirements:

2.) Thank you for including your ethics statement:

"A waiver of individual informed consent was granted through the Human Health Research Ethics Committee of Stellenbosch University (S18/10/262) and approval was granted by Tygerberg Hospital management."

Once you have amended this statement in the Methods section of the manuscript, please add the same text to the “Ethics Statement” field of the submission form (via “Edit Submission”).

3.) Please clarify whether the data utilized in this study were de-identified/anonymised before access?

4.) Please include a copy of Table 2 which you refer to in your text on page 11.

Additional Editor Comments (if provided):

Thank you for your manuscript submission to PLoS One and for giving us the opportunity to review the the manscript. The paper is well written, with good use of scientific English. Correct medical terminology has been used. Your manuscript is suitable for publication on condition that you have

Reviewers' comments:

Reviewer's Responses to Questions

**Comments to the Author**

1. Is the manuscript technically sound, and do the data support the conclusions?

Reviewer #1: Partly

Reviewer #2: Yes

Reviewer #3: Yes

2. Has the statistical analysis been performed appropriately and rigorously? 

Reviewer #1: Yes

Reviewer #2: Yes

Reviewer #3: Yes

3. Have the authors made all data underlying the findings in their manuscript fully available?

Reviewer #1: No

Reviewer #2: No

Reviewer #3: Yes

4. Is the manuscript presented in an intelligible fashion and written in standard English?

Reviewer #1: Yes

Reviewer #2: Yes

Reviewer #3: Yes

5. Review Comments to the Author

---

## [Author Response · Author response to Decision Letter 0]

9 Dec 2020

1. The basis of the recommendation to use piperacillin tazobactam + amikacin in the latter half of the study requires further explanation. Data from the first half of the study period was published in 2015 and already demonstrated a 73% ESBL rate. Was this antibiogram-directed? If not, what was the basis of selecting these two agents for the unit?

This basis of this recommendation has been included in more detail in lines 126-131. Piperacillin-tazobactam and amikacin’s activity was reviewed on all blood culture isolates for all patients at Tygerberg Hospital (including adults) in response to the emergence of CREs in the hospital in 2015. This combination displayed reasonable activity. In view of the above, and in consideration of the fact that the diagnosis of HAI is complex, that some patients may not have bacterial infection as a cause for their deterioration, and that not all patients with HAI have positive blood cultures, it was decided in consultation with Infectious Diseases (adult and paediatric), Pharmacy and the hospital’s antimicrobial stewardship committee that piperacillin-tazobactam and amikacin would be recommended as empiric therapy for HAI in patients who were stable. This was mirrored by recommendations made by the Western Cape Provincial Antimicrobial Stewardship Committee. In neonates, we maintained a low threshold to escalate to meropenem if meningitis could not be excluded, if they were unstable or if they were critically ill/admitted to ICU. 

2. The author does not differentiate between derepressed AmpC and ESBLs among S.marcescens isolates. Considering the large proportion of these isolates further detail, if available retrospectively, would be beneficial. 

Differentiation of the underlying mechanism of cephalosporin resistance in isolates with chromosomal ampC genes is imprecise, and confirmation of the molecular mechanisms of resistance in these isolates is not carried out in routine diagnostic practice. Due to the retrospective nature of this folder review, we could not access the isolates to perform further testing. This has been elaborated on in lines 116-120 and lines 161-164. We have now described these isolates as “3rd-generation cephalosporin resistant (3GCR)”, in lines 160-161. We agree with the reviewer’s suggestion that further investigation of these isolates is warranted given the large numbers, and will attempt to investigate and report on this more comprehensively in a future research study. We have also listed this as a limitation (lines 383-385).

3. The resistance phenotypes are brushed over fairly superficially. More detail can be provided in the results – specifically, what proportion of isolates were XDR and PDR. 

The resistance phenotypes have been revised to reflect ESBLs or CREs, which we felt was a more clinically useful description of the Enterobacterales group (lines 158-164). The MDR, XDR and PDR descriptive terms have been retained for the non-fermenting Gram-negative organisms. This is now outlined in lines 164-165. 

4. Which carbapenemase enzymes were detected in the isolates? Was there an outbreak of K.pneumoniae CRE in 2019 to account for the 25% CRE among K.pneumoniae in that year? 

NDM-1 metallo beta-lactamases were identified in all of the CRE isolates in the study (all 4 of which were K. pneumoniae), now reflected in lines 245-246. A CRE outbreak occurred on the neonatal platform towards the end of December 2018 (the end of the study period) and continued into 2019. The molecular investigation results of this outbreak, although not falling entirely within the time frame of the study, have been added to the results for context (lines 247-252).

5. A significant decrease in HA-BSI rates is noted in the latter half of the study period. However, these results are not discussed in the manuscript – no possible reasons for the decrease are suggested. 

This finding has been added in more detail in the discussion (lines 287-294). We initially felt that the 5-year distinction was arbitrary and that the difference we found, whilst reaching statistical significance, had unclear clinical significance. However, we agree with the reviewers’ suggestion as we have reported this result. We have also noted that the trend over the full 10-year study period was not significant in lines 194-195 (Results), although the Period 1 vs Period 2 comparison did reach significance.

6. Elaborate on why a retrospective folder review was performed on patients who died. Was it to confirm that there was no other attributable cause of death? 

The retrospective folder review was performed in the first time period (2009-2013) for a more detailed analysis of neonatal mortality in the subset who demised. The present study was not designed to comment on neonatal mortality in the same way, and we defined BSI-attributable mortality using a threshold of death within 72 hours for this analysis (lines 166-168). We have removed this statement (lines 145-146 on manuscript with tracked changes).

7. Line 83: Study design. A definition for an episode of HA-BSI should be included in the methods. Although alluded to in data collection line 124, this should be specifically stated and included in the study definitions section. 

We have added the definition in lines 149-153 for clarity of reading.

8. Investigation of suspected neonatal HA-BSI: Specify if resistance mechanisms were confirmed using other methods e.g. to differentiate between ESBL and AmpC production among Serratia marcescens.

As addressed in Comment 2: Differentiation of the underlying mechanism of cephalosporin resistance in isolates with chromosomal ampC genes is imprecise, and confirmation of the molecular mechanisms of resistance in these (and other) isolates is not carried out in routine diagnostic practice. Due to the retrospective nature of this folder review, we could not access the isolates to perform further testing. This has been elaborated on in lines 116-120 and lines 161-164. We have now described these isolates as “3rd-generation cephalosporin resistant (3GCR)”, in lines 160-161. We agree with the reviewer’s suggestion that further investigation of these isolates is warranted given the large numbers, and will attempt to investigate and report on this more comprehensively in a future research study. We have also listed the lack of additional molecular testing of resistance mechanisms as a limitation (lines 383-385).

9. Line 116: Management of suspected neonatal HA- BSI. Explain the basis of this recommendation – the paper referenced in this section of the manuscript by Dramowski in 2015 describes a 73% ESBL rate

As addressed in Comment 1: The basis of this recommendation has been included in more detail in lines 126-131. Piperacillin-tazobactam and amikacin’s activity was reviewed on all blood culture isolates for all patients at Tygerberg Hospital (including adults) in response to the emergence of CREs in the hospital in 2015. This combination displayed reasonable activity. In view of the above, and in consideration of the fact that the diagnosis of HAI is complex, that some patients may not have bacterial infection as a cause for their deterioration, and that not all patients with HAI have positive blood cultures, it was decided in consultation with Infectious Diseases (adult and paediatric), Pharmacy and the hospital’s antimicrobial stewardship committee that piperacillin-tazobactam and amikacin would be recommended as empiric therapy for HAI in patients who were stable. This was mirrored by recommendations made by the Western Cape Provincial Antimicrobial Stewardship Committee. In neonates, we maintained a low threshold to escalate to meropenem if meningitis could not be excluded, if they were unstable or if they were critically ill/admitted to ICU.

10. Line 126: Data collection. Please clarify if 100 days was chosen at random or if this was the maximum number of days at which HA-BSI was observed.

The 100-day threshold was chosen with a logical theoretical basis, in view of the fact that many preterm neonates remain hospitalised for up to 3 months. We included a small buffer margin for patients with very prolonged hospital stays, and felt that this cut-off was most suitable of the population we wanted to study. An earlier age cut-off would exclude some infants (and therefore lead to an underestimate of HA-BSI) and a later date would lead to inclusion of older infants who are no longer neonates, resulting in a potential overestimation of neonatal HA-BSI. We have expanded on this in lines 150-153.

11. Lne 209: in the Methods section, study definitions are provided including XDR and pandrug resistance. However, there is no comment on the proportion of isolates which were XDR or PDR in the results. 

As addressed in Comment 3: The resistance phenotypes have been revised to reflect ESBLs or CREs, which we felt was a more clinically useful description of the Enterobacterales group (lines 158-164). The MDR, XDR and PDR descriptive terms have been retained for the non-fermenting Gram-negative organisms. This is now outlined in lines 164-165.

12. Figure 2: Considering the larger proportion of AmpC producing S.marcescens isolates presented, was it possible to determine which isolates were true ESBL’s and which were derepressed AmpC’s? If this distinction cannot be made due to the nature of data collection, then this should be stated in the methods section, and a limitation added to the discussion. 

As addressed in Comment 2: Differentiation of the underlying mechanism of cephalosporin resistance in isolates with chromosomal ampC genes is imprecise, and confirmation of the molecular mechanisms of resistance in these isolates is not carried out in routine diagnostic practice. Due to the retrospective nature of this folder review, we could not access the isolates to perform further testing. This has been elaborated on in lines 116-120 and lines 161-164. We have now described these isolates as “3rd-generation cephalosporin resistant (3GCR)”, in lines 160-161. We agree with the reviewer’s suggestion that further investigation of these isolates is warranted given the large numbers, and will attempt to investigate and report on this more comprehensively in a future research study. We have also listed this as a limitation (lines 383-385).

13. Carbapenem resistant Enterobacterales and Acinetobacter baumannii species are on the WHO Priority Pathogen’s Critical List. Additional details regarding these in the results and discussion would increase the impact of the manuscript. 

Also see Comment 4. Additional detail has been added for CREs to lines 242-245, and more emphasis has been placed on CREs in the discussion of the neonatal outbreak (lines 246-251, lines 331-335). We have discussed the prominence of A. baumannii in lines 309-313 and have referenced a publication on this pathogen from our unit.

14. Figure 2: There appears to have been a change in epidemiology of CRE’s during period 1 and Period 2. It would be useful to elaborate, if possible, on the change. Were E.coli CRE’s due to the same carbapenemases enzyme as K.pneumoniae CRE’s in the latter part of the study?

As addressed in Comment 4: Figure 2 has also been corrected in response to the review of the. NDM-1 metallo beta-lactamases were identified in all of the CRE isolates in the study (all 4 of which were K. pneumoniae), now reflected in lines 244-245. A CRE outbreak occurred on the neonatal platform towards the end of December 2018 (the end of the study period) and continued into 2019. The molecular investigation results of this outbreak, although not falling entirely within the time frame of the study, have been added to the results for context (lines 246-251).

15. In the discussion, the authors mention a personal communication that 25% of K.pneumoniae isolates in 2019 were carbapenem resistant. However, this data is not reflected in this section of the results. Did this finding correspond to a documented outbreak in the unit?

This has been discussed in more detail in lines 246-251 (Results) and lines 331-335 (Discussion). A documented CRE outbreak did occur in the unit with the first cases identified in the last few weeks of December 2018. The outbreak has been investigated with the results forming part of a PhD study currently in progress (not yet published). Given the ongoing nature of the outbreak investigation, and the fact that the outbreak fell outside the period of this study, we have provided only limited additional information about the outbreak.

16. In the discussion: A significant decrease in HA- BSI rate is reported, however the author does not offer possible reasons for this reduction

As addressed in Comment 5: This finding has been added in more detail in the discussion (lines 286-293). We initially felt that the 5-year distinction was arbitrary and that the difference we found, whilst reaching statistical significance, had unclear clinical significance. However, we agree with the reviewers’ suggestion as we have reported this result. We have also noted that the trend over the full 10-year study period was not significant in lines 194-195 (Results), although the Period 1 vs Period 2 comparison did reach significance.

17. Line 279: The comment about reduced virulence of circulating strains should be referenced 

This has been added (line 321).

18. Line 287: As per comment in results section. Over the 10- year study period, could changes in CLSI interpretive criteria have affected resistance reporting rates for ESBL’s and CRE’s? If so, these need to be discussed. 

Thank you for highlighting this omission. We agree that it is crucial to include this information in studies such as this, and have included a brief summary of the relevant changes and a comprehensive reference (lines 143-146, lines 327-331). We have also discussed these changes under the limitations (lines 385-390).

19. Conclusion: In addition to surveillance, the value of unit antibiogram provision to review empiric recommendations should be included. 

This has been included in the conclusion (lines 403-405).

20. Line 37: Abstract Sentence amendment: " The crude mortality rate of neonatal HA-BSI increased from Period 1 to Period 2 from 139/717 (19.4%) to 179/718 (24.9%) 

Completed as recommended, line 38.

21. Line 60: remove comma before "and inadequate reporting”, line 124, use " in a neonatal ward" in place of "on", line 286, use " but have emerged in the neonatal unit" in place of "on". 

Completed as recommended.

22. Line 129: suggest reword for clarity: Retrospective folder review was conducted for patients who died to determine if the case would be categorized as attributable or crude mortality. 

Also see Comment 6. This statement has subsequently been removed.

23. Line 242: Where the term “cover” is used to imply in vitro activity of an antibacterial, this should be changed to “antibacterial activity for empiric use” 

Completed throughout the manuscript as recommended.

24. Line 246: Discussion. Consider rewording “shift in draining patterns” to allow for improved understanding by an international readership – consider “changes to defined regional populations serviced by hospitals” 

Completed as recommended (line 281).

25. Line 246: Add “increased” urbanization 

Completed as recommended.

26. Line 248: There were “fewer” blood cultures submitted 

Completed as recommended.

27. References: Please check references The reference list appears twice 

Second list removed.

28. Journal title incorrectly captured by reference manager- reference 13 

This has been rectified.

---

## [Editor Report · Decision Letter 1]

22 Dec 2020

A retrospective analysis of pathogen profile, antimicrobial resistance and mortality in neonatal hospital-acquired bloodstream infections from 2009-2018 at Tygerberg Hospital, South Africa

PONE-D-20-25473R1

Dear Dr. Reddy

We’re pleased to inform you that your manuscript has been judged scientifically suitable for publication and will be formally accepted for publication once it meets all outstanding technical requirements.

Kind regards,

Adriano Gianmaria Duse, MD

Academic Editor

PLOS ONE
---

## [Editor Report · Acceptance letter]

5 Jan 2021

PONE-D-20-25473R1 

A retrospective analysis of pathogen profile, antimicrobial resistance and mortality in neonatal hospital-acquired bloodstream infections from 2009-2018 at Tygerberg Hospital, South Africa 

Dear Dr. Reddy:

I'm pleased to inform you that your manuscript has been deemed suitable for publication in PLOS ONE. Congratulations! Your manuscript is now with our production department. 

Kind regards, 

on behalf of

Dr. Adriano Gianmaria Duse 

Academic Editor

PLOS ONE